# The Challenges of Partnering to Promote Health through Sport

**DOI:** 10.3390/ijerph18137193

**Published:** 2021-07-05

**Authors:** Alex Donaldson, Kiera Staley, Matthew Cameron, Sarah Dowling, Erica Randle, Paul O’Halloran, Nicola McNeil, Arthur Stukas, Matthew Nicholson

**Affiliations:** 1Centre for Sport and Social Impact, La Trobe Business School, La Trobe University, Bundoora, VIC 3086, Australia; k.staley@latrobe.edu.au (K.S.); e.randle@latrobe.edu.au (E.R.); p.ohalloran@latrobe.edu.au (P.O.); n.mcneil@latrobe.edu.au (N.M.); A.Stukas@latrobe.edu.au (A.S.); m.nicholson@latrobe.edu.au (M.N.); 2Victorian Health Promotion Foundation (VicHealth), West Melbourne, VIC 3003, Australia; mcameron@vichealth.vic.gov.au (M.C.); sdowling@vichealth.vic.gov.au (S.D.); 3School of Psychology and Public Health, La Trobe University, Bundoora, VIC 3086, Australia; 4Monash University Malaysia, Subang Jaya 47500, Malaysia

**Keywords:** settings-based health promotion, community sports clubs, partnerships, concept mapping, collaborations, physical activity

## Abstract

Interagency partnerships and collaborations underpin a settings-based approach to health promotion in all settings, including sport. This study used an online concept mapping approach to explore the challenges that Regional Sports Assemblies (RSAs) in Victoria, Australia experienced when working in partnerships to develop and deliver physical activity programs in a community sport context. Participants from nine RSAs brainstormed 46 unique partnership-related challenges that they then sorted into groups based on similarity of meaning and rated for importance and capacity to manage (6-point scale; 0 = least, 5 = most). A six cluster map (number of statements in cluster, mean cluster importance and capacity ratings)—Co-design for regional areas (4, 4.22, 2.51); Financial resources (3, 4.00, 2.32); Localised delivery challenges (4, 3.72, 2.33); Challenges implementing existing State Sporting Association (SSA) products (9, 3.58, 2.23); Working with clubs (8, 3.43, 2.99); and Partnership engagement (18, 3.23, 2.95)—was considered the most appropriate interpretation of the sorted data. The most important challenge was Lack of volunteer time (4.56). Partnerships to implement health promotion initiatives in sports settings involve multiple challenges, particularly for regional sport organisations working in partnership with community sport clubs with limited human and financial resources, to implement programs developed by national or state-based organisations.

## 1. Introduction

Cross-sector partnerships, coalitions and alliances are critical to collective action to promote health [1], particularly physical activity [2], at local [3] and national [2] levels. A partnership in the public health context has been defined as “where two or more parties commit to work together for a common purpose” [4]. In an ideal world, partnerships use a diversity of skills and resources from multiple organisations and agencies for better health promotion outcomes [5].

Building, leveraging and sustaining partnerships underpins all settings-based health promotion work [6]. Whether establishing healthy cities [7], universities [8], schools [9], health services [10], prisons [11], workplaces [12] or sports stadia [6], partnerships are a key strategy for success. Within the health promoting sports club literature, best-practice policy development guidelines include collaborating with other clubs [13]. A systematic mapping of how the settings-based approach was applied through health promotion interventions in sports clubs included collaboration with external agencies, such as health agencies, via building common cultures and developing agreed collaboration processes [14]. Partnerships are also a key strategy underpinning an evidence-driven intervention framework for planning, developing and implementing health promotion initiatives in sports club settings [15]. In the process of developing this framework, published indicators for assessing the health promotion status of sports clubs were reviewed and “involving partners and the community in health promotion actions and decision making” emerged as one of five indicators of a health promoting sports club [15]. Partnerships are also a key component of manuals, checklists and best-practice guidelines to assist community sports clubs to promote health [16,17].

Despite the strong theoretical and conceptual arguments for partnerships as a key pillar to developing a settings-based approach to health promotion in sport settings, there is little empirical research exploring the nature of such partnerships or how they work [18]. In a series of studies published between 2009 and 2011, Casey and colleagues explored the partnership development and implementation processes associated with successful sport and recreation programs [19,20]. They concluded that successful partnerships require long-term commitments, in addition to trust, shared interests and open communication between partners [19]. Partnerships most likely to succeed were: led by sport and recreation organisations; engaged key stakeholders; had a diversity of skills, resources and approaches; developed formal partnership agreements; and adopted phased approaches to program development and implementation [20]. More recently, research into a Canadian public health–academic–sport and recreation partnership aimed at increasing physical activity identified several factors constraining the implementation of the partnership [18]. These included organisational misalignments in capacity and readiness to collaborate, power imbalances, and differences in language, norms and values [18]. A study of health promotion in French sports clubs concluded that coaches perceived that partnerships were the least implemented dimension of a health promoting sports club and also the least relevant health promoting dimension for coaches [21].

Whilst the available research focuses on the hallmarks of effective partnerships, relatively little is known about the challenges of sporting organisations working together to enhance and deliver health promotion activities for the communities they serve. Indeed, a review of international health promoting sports club research concluded that more work is needed to better understand how partnerships can be leveraged to encourage health promotion action in sporting organisations [22].

In 1983, the Victorian State Government established Regional Sports Assemblies (RSAs) following the release of the ‘Sport in Victoria’ policy paper on the future planning, development and funding for sport. The nine RSAs support sport and recreation groups within their regional catchment through direct support and advice to clubs, support the roll out of state and national projects, and build networks and partnerships within the sport and recreation sector. Readers can find out more about RSAs at https://www.regionalsportvictoria.org.au/about-us/ (accessed on 1 July 2021).

The Victorian Health Promotion Foundation (VicHealth) sponsors sporting and cultural activities. It works with partners, including community sport clubs, RSAs and state-wide sport governing bodies, to promote health, primarily through a settings-based approach to health promotion [23]. VicHealth partnered with the nine RSAs in the Regional Sport Program to use innovative programs and initiatives to promote physical activity through sport in rural and regional areas from 2015 to 2021. Recognising that no sector can address the complexity of increasing sport participation working alone [24], VicHealth funded the RSAs to work with local partners, including community sport clubs, local government authorities, health services, sport governing bodies and community support services, to create more opportunities for Victorians to be physically active in sport and active recreation settings. RSAs were asked to develop locally relevant strategies informed by community consultation and local insights, gaps and opportunities. Based on these strategies, RSAs worked with local and state partners to initiate and support delivery of participation opportunities, and support clubs and other stakeholders to ensure successful programs were sustainable.

Examples of the roles of RSAs in the Regional Sport Program included: (1) expanding a social sport product developed by a State Sporting Association (SSA) into regional Victoria; and (2) establishing a social sport program for regional school children in a community club. Within these roles, the RSAs: deliver programs; provide feedback to SSAs on program delivery and cost; identify potential program deliverers; contribute to customised marketing materials for regional needs; support community clubs to promote and deliver programs; assist SSAs and clubs to deliver local marketing; engage other stakeholders where relevant (e.g., local councils to access facilities); and assist with data management and evaluation.

Figure 1 depicts the complex network of organisations that deliver the Regional Sport Program. Not all links are relevant to every RSA and the strength of the link both in the number of partners and the nature of the partnership varied depending on the specifics of the project.

This research explores the partnership-related challenges that the RSAs in Victoria, Australia experienced while working in partnerships to create more opportunities for Victorians to be physically active in sport and active recreation settings.

## 2. Methods

Given the exploratory nature of this research, we used Concept Mapping (CM), a mixed-method participatory approach that is well suited to developing conceptual frameworks of complex topics [25,26]. The ways in which we applied the key CM steps of preparation, ideas generation (brainstorming), statement structuring (sorting and rating) and analysis [27], are outlined in Figure 2. We used the Concept Systems Global Max™ [28] web platform to undertake all aspects of this study.

### 2.1. Sample Selection and Recruitment

In October 2018, we invited multiple contacts (total *n* = 35: range = 2–6 per organisation) from each of the nine RSAs funded through the Regional Sport Program, to participate in the CM exercise. The emailed invitations included a hyperlink to the brainstorming activity. We sent several reminder emails to all potential participants before the idea generation step closed after 14 days.

Before undertaking their first CM activity, participants were asked to describe their position in their organisation and their length of employment with their organisation. These background questions were categorical with multiple choice responses. Participants were also asked to rate their level of experience and confidence engaging in and managing the partnerships necessary to develop and deliver physical activity programs for less active people (i.e., people who do not meet the Australian Physical Activity Guidelines) [29]. Both these ratings were on a 6-point scale from 0 (no experience/not at all confident) to 5 (lots of experience/very confident).

### 2.2. Data Collection

The brainstorming question used to generate ideas in this study was: “What partnership-related challenges has your organisation experienced when developing and delivering physical activity programs for less active people?”. The two rating instructions used were: “On a scale from 0 (least important) to 5 (most important), how important is this challenge to developing and delivering physical activity programs for less active people?” and “On a scale from 0 (least capacity) to 5 (most capacity), what capacity does your organisation have to manage this challenge to developing and delivering physical activity programs for less active people?”. We asked participants to brainstorm as many single-thought statements as they could in response to the brainstorming question. Participants could review the statements other participants made and access the online platform multiple times [27].

Members of the research team (authors AD, KS, SD and MC) conducted multiple rounds of synthesising and editing the brainstormed statements to: delete statements unrelated to the focus prompt; split compound statements; identify statements that represented the same idea and select the statement that best captured the essence of the idea; and edit statements (if necessary) to reflect an agreed meaning [27]. Every effort was made to ensure statements reflected the original voice of the participant where possible. This iterative process continued until there was consensus that the final statement list contained a manageable set of unique, clear and pertinent ideas. We cross-referenced the final list to the original lists of statements to ensure all relevant brainstormed ideas were represented in the final list.

We invited all the originally identified RSA representatives (*n* = 35) to participate in the statement structuring, even if they had not participated in the brainstorming. Multiple reminder emails were sent to anyone who had not responded or completed the sorting and rating tasks over two weeks in December 2018.

During the statement structuring process, each participant sorted the randomized synthesised statements into groups that made sense to them. They were instructed to group statements according to similarity in meaning, and to name each group based on its theme or contents. Participants could create single-statement groups if they thought a statement was unrelated to all other statements. They were asked to put every statement somewhere, and to avoid creating ‘miscellaneous’ or ‘other’ groups. They were also informed that 5 to 20 groups usually work well to organise the number of statements they were asked to sort [30]. Participants were also instructed to rate each statement on ‘importance’ and ‘capacity to manage’, using the full six-point scales (0–5), relative to the other challenges in the list.

### 2.3. Data Analysis

During the multi-stage data analysis, we created a square symmetric similarity matrix from the sorted data, before applying two-dimensional non-metric multidimensional scaling to locate each statement as a separate point on a two-dimensional X-Y ‘point map’. We then used hierarchical cluster analysis to partition the point map into groups of statements creating a ‘cluster map’. A detailed description of the multidimensional scaling, including the stress index calculation, and hierarchical cluster analysis used in the Concept Systems Global MAX™ [28] web platform, is available from Kane and Trochim (pp. 87–100, [27]). We also calculated mean importance and capacity to manage ratings for each statement, and used them to generate a “go-zone” graph, in which we plotted each statement’s mean ratings on a graph divided into four quadrants using the grand mean of each rating as the axes. We calculated a correlation coefficient to demonstrate the degree of linear association between the two rating variables.

To select the most appropriate number of clusters, we followed Kane and Trochim’s recommended process (pp. 101–103, [27]). We examined the cluster maps for a 9-cluster solution through to a 5-cluster solution, paying particular attention to which clusters of statements were merged as the number of clusters decreased. This negotiated process was used to identify the cluster level that the research team believed retained the most useful detail between clusters, while merging those clusters that seemed to logically belong together. After we agreed on the most appropriate cluster level, we identified any statements that subjectively seemed to belong in an adjacent cluster and re-drew the cluster boundaries so they were included in the more appropriate neighbouring cluster [31].

## 3. Results

Thirty-one individual participants from all nine of the funded RSAs contributed CM data—30 in the idea generation, 26 in the statement sorting, 27 in the importance rating, and 24 in the capacity to manage rating. Twenty-four participants contributed data in all phases, while four contributed to the idea generation only.

Thirty participants provided background information about themselves when they first contributed data to the CM process. Half of participants described their position as a program coordinator (*n* = 15, 50%), while 20% (*n* = 6) described their position as an executive officer. Half (*n* = 15, 50%) of the participants had been employed with their current organisation for three years or longer. The mean ratings for participants’ experience and confidence in engaging in and managing the partnerships necessary to develop and deliver physical activity programs for less active people were 3.62 (range 0–5; Standard Deviation (SD) = 1.42) and 3.82 out of 5 (range 1–5; SD = 0.17), respectively. Full details of the participants’ responses to the background questions can be found in Table 1.

The participants brainstormed 73 challenges in response to the project focus prompt. The research team synthesised and edited these down to 46 unique challenges for participants to sort and rate (Table 2). Twenty-six participants sorted the 46 challenges into groups (mean = 7.5 groups; mode = 7 groups (11 participants); range = 4–13 groups).

The grand mean importance rating for all challenges was 3.51 out of 5 (SD = 0.53) (Table 2). Challenges in the *Co-design for regional areas* cluster were rated the most important (4.22; SD = 0.10) while those in the *Partnership engagement* cluster were rated the least important (3.23; SD = 0.40). The grand mean capacity to manage rating for all challenges was 2.68. Participants rated their capacity to manage the challenges in the *Working with clubs* cluster the highest (2.99; SD = 0.15), and those in the *Challenges implementing existing SSA products* cluster the lowest (2.23; SD = 0.40).

The individual challenge with the highest mean importance rating was #27 *Lack of volunteer time* (in the *Localised delivery challenges* cluster) with a mean importance rating of 4.56 (SD = 0.75). The individual challenge with the lowest mean importance rating was #32 *Challenges related to developing promotional materials for programs (e.g., cooperation)* (in the *Partnership engagement* cluster) with a mean importance rating of 2.48 (SD = 1.19). The individual challenge with the highest mean capacity to manage rating was #42 *Difficult to find good partners in regional/rural area* (in the *Partnership engagement* cluster) with a mean capacity to manage rating of 3.71 (SD = 0.95). The individual challenges with the lowest mean capacity to manage rating were #27 *Lack of volunteer time* (in the *Localised delivery challenges* cluster) and #45 *Distance for SSAs to travel to rural and remote areas* (in the *Challenges implementing existing SSA products* cluster), both with a mean capacity rating of 1.67 (SD = 1.09 and 1.40, respectively).

The research team agreed that a 6-cluster solution—encompassing *Co-design for regional areas* (4 challenges); *Financial resources* (3 challenges); *Localised delivery challenges* (4 challenges); *Challenges implementing existing SSA products* (9 challenges); *Working with clubs* (8 challenges); and *Partnership engagement* (18 challenges)—retained the most useful detail while merging those clusters that seemed to conceptually belong together (see Figure 3). The distances between the points on the cluster map (Figure 3) represent the degree of perceived similarity between challenges (i.e., the challenges grouped together by more participants are located closer to each other on the map). For example, challenges #7 and #41 were considered so closely related that 23 out of 26 of participants grouped them together. By contrast, challenges #5 and #28 were considered so unrelated that no participants grouped them together. The stress index—an indication of how well the two-dimensional map reflects the square symmetric similarity matrix generated from the sorted data—was 0.22, close to the average stress value across a broad range of CM projects [30]. A full list of the challenges within each cluster, including the six challenges that were reassigned to neighbouring clusters to which there was a better conceptual fit is provided in Table 2.

Figure 4 is a go-zone graph for all 46 challenges. The top right “go-zone” quadrant of challenges contains the eight challenges rated above average on both importance and capacity to manage. The Pearson Product Moment Correlation coefficient indicates moderate correlation between ‘Importance’ and ‘Capacity to address’ (r = −0.49) [32]. To aid interpretation of the go-zone graph, see Table 2 for the details of each challenge, including its mean importance and capacity to manage ratings.

## 4. Discussion

This research explores the partnership-related challenges that the RSAs in Victoria, Australia experienced while working in partnerships to create more opportunities for Victorians to be physically active in sport and active recreation settings. The findings add to the limited body of literature about partnerships to promote health within sports club settings. Importantly, this study highlights some of the structural barriers to developing and delivering physical activity programs via a partnership model, especially in regional and rural areas. Foremost among these barriers is the instrumental nature of sports clubs, combined with the pressure on a limited pool of over-stretched volunteers. The RSAs in this study identified a lack of volunteer time as one of the major partnership-related challenges, making it difficult to establish a common purpose. This was further compromised by the mismatch between physical activity promotion programs which are the core business of health promotion agencies such as VicHealth, and the raison d’etre of community sports clubs, which is to field competitive teams. The lack of volunteer time and frequently changing personnel also makes it difficult for community sport clubs to establish consistent communication and high levels of commitment to health promotion partnerships and projects. These fundamental challenges, of time, purpose, communication and commitment, are exacerbated by the regional and rural locations of the sports clubs and RSAs, which in Australia is more significant because of the physical distance between urban and regional areas. In this study, it appears that using community sport and recreation settings to provide physical activity programs in regional areas, where they are often most needed, has additional challenges that exacerbate the already significant challenges of forming partnerships to promote health through sport clubs.

It is clear from the findings of this study of a sports-setting based partnership to deliver physical activity opportunities, that establishing a common purpose or goal for the partnership was a major challenge. The statements in the *Co-design for regional areas* cluster directly relate to communication and collaboration challenges between the SSAs responsible for developing physical activity products or programs and the RSAs, community sports clubs and local government authorities with a comprehensive understanding of the implementation context (i.e., the sports club setting context), at the initiation of the partnership. It is highly likely that these challenges were amplified by the fact that the SSAs needed to establish partnerships with multiple RSAs to support the delivery of their innovative physical activity programs across the state while, simultaneously, RSAs had partnerships with multiple SSAs to support the delivery of a wide range of products in their region. This finding is supported by Butterfoss and colleagues’ notion that articulating a common purpose is the most important element in coalition formation and can be facilitated by open communication [33].

All the challenges in the *Working with clubs cluster*—particularly those located in quadrant 1 of the go-zone, related to working with clubs to promote physical activity when they are already consumed with core business—reflect a lack of synergy between the focus of the partnership (i.e., getting people more active) and the primary purpose of community sports clubs (i.e., delivering competitive sport opportunities for members). This highlights the importance of building a common culture and goal for collaborating to promote health in sports club settings [14]. It also supports earlier sports-setting health promotion partnership research that identified differences between the aims and strategic priorities of professional organisations (such as SSAs and RSAs) and volunteer-run community sporting clubs [19].

Successful health promotion partnerships and coalitions are underpinned by the strong commitment of individual member agencies [33]. Our study and other similar studies [19] highlight some of the challenges in gaining and sustaining commitment from partner agencies to promote physical activity in sports club settings. For example, from the perspective of the RSAs that participated in this concept mapping study, finding committed people in community sports clubs, partners not delivering what they promised and different levels of motivation and commitment from partner organisations, all emerged as relatively important partnership formation and implementation challenges. Shared decision making can facilitate commitment to the health promotion partnerships [33] and our findings suggest that engaging all partner agencies to co-design health promotion interventions to ensure contextual relevance may be one avenue to build commitment to partnerships to promote health in sports club settings.

The limited financial, human and organisational system resources among all agencies involved in the partnership we studied were important challenges that emerged from our study. The lack of capacity of community sporting organisations to embrace health promotion initiatives, the lack of staff and financial resources at the RSAs, SSAs and community clubs, and the inadequacy of program funding models were all identified as challenges to working in partnership to promote physical activity through sports clubs. This is unsurprising, as having access to adequate financial, human and capacity building resources to invest in health promotion is a key strategy within the health promoting sports club model [14]. A lack of agency capacity has also previously been identified as a limiting factor to implementing and sustaining multi-agency health promotion initiatives in sports club setting [19,34].

The moderate negative correlation between the RSAs’ perceptions of the importance of the challenges to working in partnership to develop and deliver physical activity programs and their self-rated capacity to manage the challenges suggests that there are some relatively important partnership-related challenges that the RSAs do not believe they have a great capacity to manage. More specifically, the two challenges rated as most important—*lack of volunteer time* and *communication with SSAs who are driving products in regional areas without engaging local clubs or RSAs/LGAs*—were both rated below the grand mean for capacity to manage. In addition, 13 of the 46 challenges and at least one challenge from every cluster except the *Working with clubs* cluster, were located in quadrant 2 of the go-zone. This finding supports previous calls for further research to better understand how to build the capacity of all agencies involved in partnerships to ensure ongoing and sustainable physical activity promotion in sports club settings [35].

An important limitation of this study is that the partnership-related challenges to promoting health in sports club settings were only identified from the perspective of one type of organisation (RSAs) in a complex, multi-agency partnership in the context of Australian sports. It is highly likely that different challenges would emerge if the same study was undertaken with other types of organisations engaged in this partnership, such as SSAs and community sports clubs, or in other sports systems. Nonetheless, the findings of this study align with the partnership-related components of the recently published health promoting sports club model [15] and guidelines [14]. They are also supported by the finding of previous health promoting sports club [19,20,34,35] and more general health promotion partnership research [33].

## 5. Conclusions

This study confirms that there are multiple and complex challenges to establishing, implementing and maintaining interagency partnerships to promote health within community sports club settings. Furthermore, it supports the call to appropriately resource and build the capacity and systems of all partnering agencies, as well as to understand the local and regional implementation context and program delivery challenges. Engaging local agencies and sports clubs in co-design, subject to adequate resourcing, may be an effective way of enhancing the contextual relevance of health promotion programs and interventions targeting sports club settings. Finally, investing time negotiating and developing a shared vision and guiding purpose for health promotion partnerships in sports club settings is highly recommended to ensure clarity between partners and to maximise the return on investments of all agencies involved.

## Figures and Tables

**Figure 1 ijerph-18-07193-f001:**
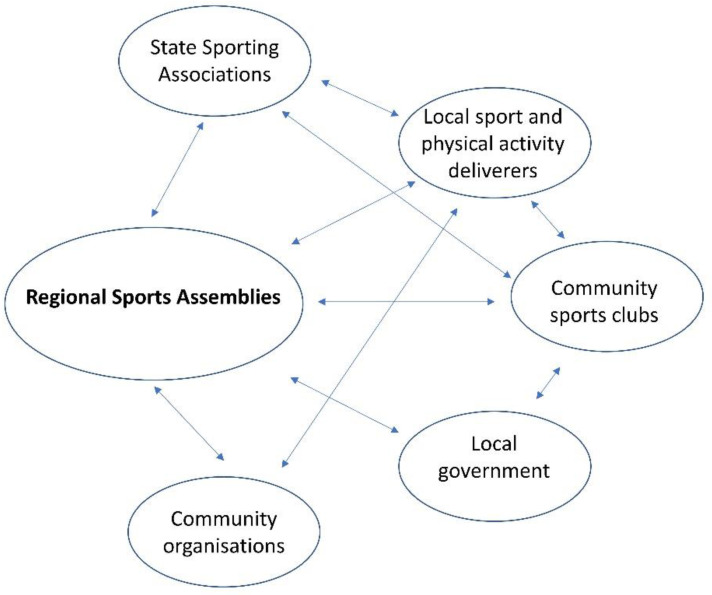
The network of organisations that partner to deliver the Regional Sport Program.

**Figure 2 ijerph-18-07193-f002:**
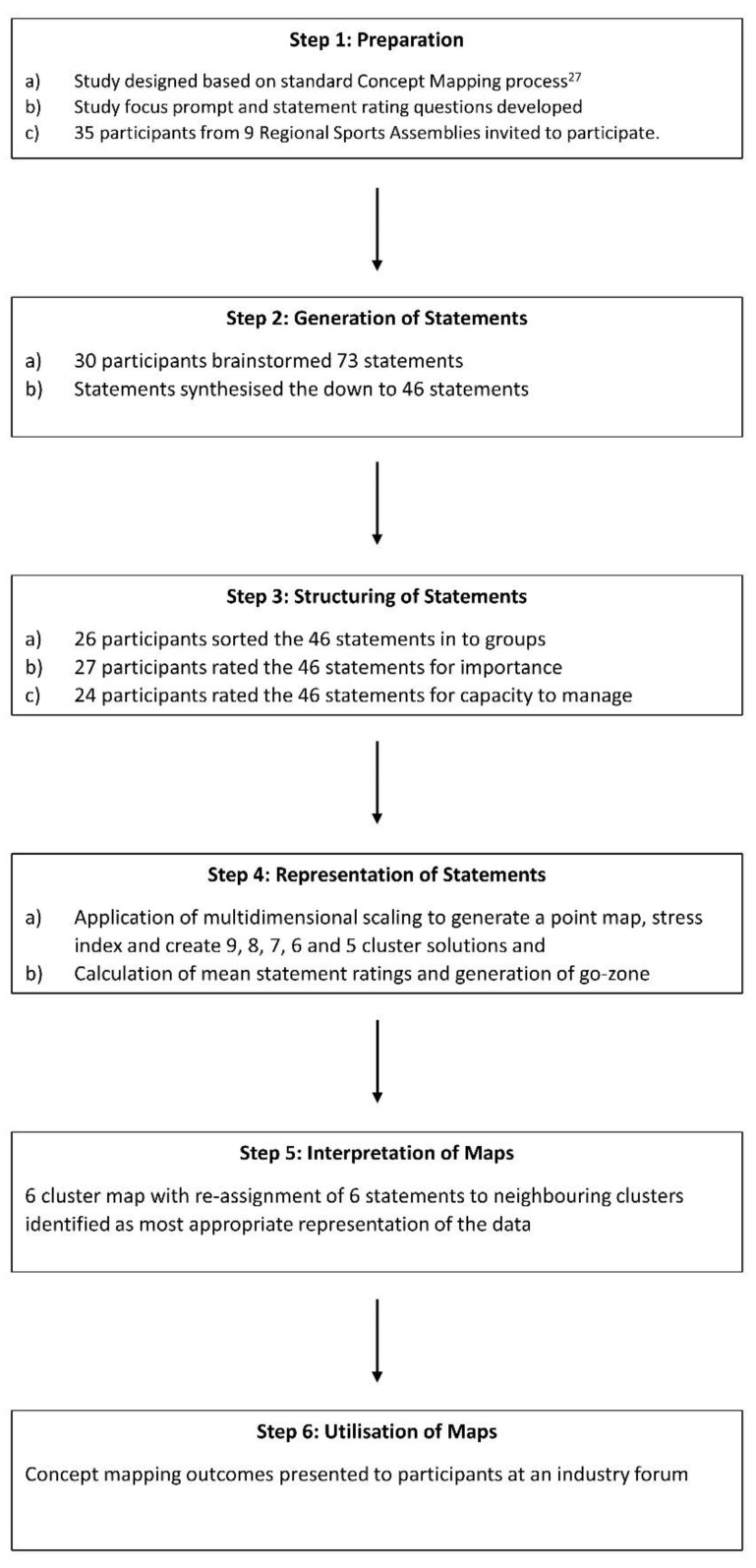
Flow of concept mapping process for this study.

**Figure 3 ijerph-18-07193-f003:**
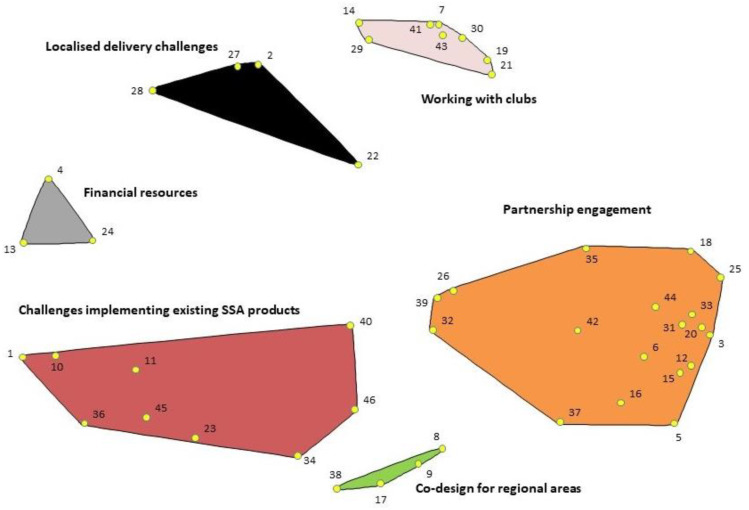
6 Cluster map of challenges to working in partnership to promote physical activity in community sport settings.

**Figure 4 ijerph-18-07193-f004:**
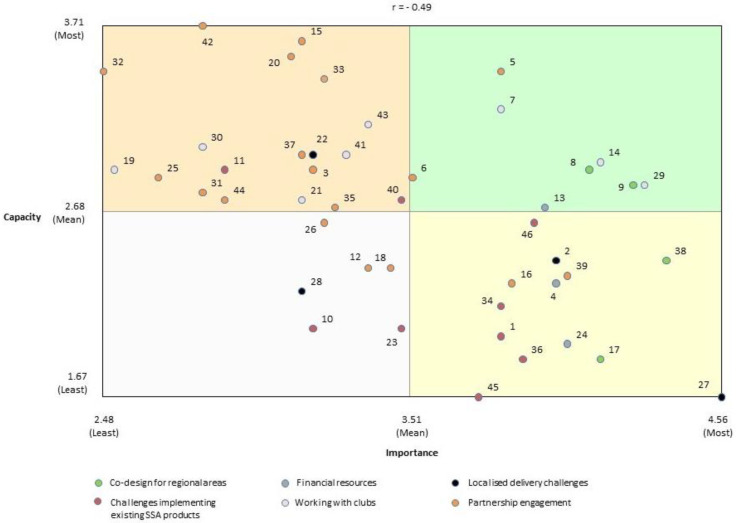
Go-zone graph of challenges to working in partnership to promote physical activity in community sport settings.

**Table 1 ijerph-18-07193-t001:** Characteristics of participants (*n* = 31).

**Organisational Position** **(*n* = 30)**	Program coordinator			15 (50%)		
	Executive Officer			6 (20%)		
	Support staff			2 (7%)		
	Other *			7 (23%)		
**Length of Employment with Current Organisation (*n* = 30)**	0–6 months			6 (19%)		
	6–12 months			3 (10%)		
	1–2 years			6 (19%)		
	3–5 years			3 (10%)		
	More than 5 years			12 (39%)		
**Confidence to Engage in and Manage Partnerships** **(*n* = 29; mean = 3.62; SD = 1.42)**	0 (not confident)	1	2	3	4	5 (very confident)
	0 (0%)	1 (3%)	1 (3%)	6 (19%)	15 (48%)	6 (19%)
**Experience Engaging in and Managing the Partnerships** **(*n* = 29; mean = 3.82; SD = 0.17)**	0 (no experience)	1	2	3	4	5 (very experienced)
	1 (3%)	2 (7%)	3 (10%)	5 (17%)	8 (28%)	10 (35%)

* Other included: Programs Manager; Team Leader; Project Officer x 3; General Manager; Project Coordinator.

**Table 2 ijerph-18-07193-t002:** Regional Sports Assembly partnership-related challenges generated during the Concept Mapping brainstorming process including the cluster in which each statement fits, mean importance and capacity to implement ratings for each statement.

Cluster and Statements		Mean Rating ^^^		*Go-Zone* *Quadrant*
		*Importance of the Challenge ^θ^*	*Capacity to Manage the Challenge ^θ^*	
**Co-Design for Regional Areas**		**4.22 ^◊^**	**2.51 ^◊^**	
38	Communication with SSAs who are driving products in regional areas without engaging local clubs or RSA/s/LGAs.	4.37	2.42	2
9	Lack of formal collaboration between an SSA and RSA in developing a strategy to implement a product in a regional or rural setting.	4.26	2.83	1
17	SSAs working in the region without communication with RSA—sometimes we find out about it after the program has been developed, making it too late to collaborate.	4.15	1.87	2
8	Lack of collaboration between an SSA and RSA when developing an idea that is suitable for regional and rural audiences.	4.11	2.92	1
**Financial Resources**		**4.00 ^◊^**	**2.32 ^◊^**	
24	A lack of program funds can restrict/limit what can be delivered between partners.	4.04	1.96	2
4	The cost in running certain programs especially to the rural audience.	4.00	2.29	2
13	Inadequate resources for staffing to facilitate/initiate opportunities.	3.96	2.71	1
**Localised Delivery Challenges**		**3.72 ^◊^**	**2.33 ^◊^**	
27	Lack of volunteer time.	4.56	1.67	2
2	Lack of resources of individual sporting clubs.	4.00	2.42	2
22	Lack of adherence to systems (e.g., registration and attendance records) or engagement in evaluation.	3.19	3.00	3
28	Lack of facilities (i.e., cannot find a space for the program to run, therefore partnership dissolves).	3.15	2.25	4
**Challenges Implementing Existing SSA Products**		**3.58 ^◊^**	**2.23 ^◊^**	
46 ^#^	Different levels of motivation and commitment applied by RSA and SSA to a program impacts success and sustainability.	3.93	2.62	2
36	Lack of flexibility in the delivery of some SSA products.	3.89	1.87	2
34 ^#^	Lack of SSA presence in the region makes it hard to effectively work together to get programs up and running.	3.81	2.17	2
1	The funding models developed by SSAs are too expensive to deliver in regional areas.	3.81	2.00	2
45	Distance for SSAs to travel to rural and remote areas.	3.74	1.67	2
40 ^#^	Distance—RSA’s covering multiple LGAs makes it difficult to meet stakeholders in person to form partnerships.	3.48	2.75	3
23	Different SSAs are at different levels of preparedness for delivering programs to less active people.	3.48	2.04	4
10	Differing views on costings (e.g., a particular SSA wanting to make a profit rather than running as a not-for-profit activity).	3.19	2.04	4
11	Reliance on a variety of SSA products divides staff focus across multiple projects and contacts.	2.89	2.92	3
**Working with Clubs**		**3.43 ^◊^**	**2.99 ^◊^**	
29	Finding the committed person/people within a club, for the potential project.	4.30	2.83	1
14	Working with clubs around developing modified products when they are already consumed with core business.	4.15	2.96	1
7	Convincing clubs about the benefits of targeting less active people	3.81	3.25	1
43	Convincing clubs targeting less active people that it will not be a lot more work.	3.37	3.17	3
41	Sporting clubs finding it hard to look through an ‘inactive lens’ and understand what this target group needs—particularly long-term.	3.30	3.00	3
21	Resistance from some sports clubs who do not want to partner, as they view it as help they do not need or want, rather than a collaboration for mutual benefit.	3.15	2.75	3
30	Differing views from clubs on what can increase participation (e.g., the benefits of a family friendly club vs. a ‘partying’ club).	2.81	3.04	3
19	Sporting club partners trying to ‘recruit’ participants to traditional sport during sessions.	2.52	2.92	3
**Partnership Engagement**		**3.23 ^◊^**	**2.95 ^◊^**	
39 ^#^	Developing strategic level partnerships takes time… time that is not often afforded to us.	4.04	2.33	2
5	Failing communication between partners.	4.00	3.46	1
16	Partners engaging us at the last minute and just being asked to promote through our networks.	3.85	2.29	2
6	Partners not always delivering on what they promised (e.g., clubs often over commit and then program is affected).	3.52	2.87	1
18	Previous program outcomes (e.g., low numbers) and program features (e.g., casual participation; child friendly, etc.) make it hard to engage new and re-engage past partners to deliver new programs.	3.44	2.37	4
12	Partner claims the project is a priority area for the organisation, but does not engage or assist.	3.37	2.37	4
35	Difficult to maintain open communication with key personnel as different projects/work/life priorities alter.	3.26	2.71	3
33	Ensuring organisations/partners are involved in the project for the right reasons.	3.22	3.42	3
26 ^#^	Clubs driving demand for programs, RSAs assisting to achieve results benefiting all stakeholders, but SSA not buying-in.	3.22	2.62	4
3	All potential partners have their own agenda, which can skew a project in line with their priorities, rather than the original intended outcome.	3.19	2.92	3
15	Understanding what the “partnership” actually is, and whether or not the allocated roles are meaningful and worthwhile.	3.15	3.62	3
37	Communication with organisations that have staff changes.	3.15	3.00	3
20	Understanding the motivations and objectives of the different partners (i.e., at club, council, RSA or SSA level).	3.11	3.54	3
44	Organisations seemingly ‘defending their turf’ and not actively working in the true spirit of partnership.	2.89	2.75	3
42	Difficult to find good partners in regional/rural area.	2.81	3.71	3
31	Differing goals for potential partners who have access to less active people (Eg LGAs youth services focusing on arts and culture, and do not want to invest or partner in sport or active recreation).	2.81	2.79	3
25	Community organisations can often act as ‘gatekeepers’ to less active people (e.g., people with a disability, multicultural communities etc) which can present some challenges in terms of participant engagement.	2.67	2.87	3
32 ^#^	Challenges related to developing promotional materials for programs (e.g., cooperation)	2.48	3.46	3
**Grand Mean for All Statements**		**3.51**	**2.68**	

^θ^ 0 (least important/least capacity to manage) to 5 (most important/most capacity to manage); ^ 27 participants rated all 46 statements for importance and 24 participants rated all statements for capacity to manage; ^◊^ mean importance/capacity to manage rating for all the statements in the cluster; ^#^ re-assigned from Co-design for regional areas cluster.

## Data Availability

The authors confirm that most of the data supporting the findings of this study are available within the article. Raw numerical data in the form of a similarity matrix and descriptive statistics were generated and may be made available in a de-identified format on request.

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
