# Peer review of "The Challenges of Partnering to Promote Health through Sport"

_ijerph, 2021, doi:10.3390/ijerph18137193_

Round 1
Reviewer 1 Report
The article “The challenges of partnering to promote health through sport” is very interesting and presents an important theme for the specific area. As a contribution to a better understanding of the work, some changes need to be made: 1. The authors separate “Introduction” and “Research context”, however, some points written in the “Research context” can be deleted, as they are not directly linked to the main topic of study, but to the historical context. In this way, I suggest that the authors summarize the information and remove the topic “Research context”, adding the short text to the “introduction”. 2. Remove “Box 1”. I suggest that the information that is in “Box 1” be written in text format, in its proper place. 3. The resolution of figures 1 and 2 is very bad in the file, which prevents reading of some important parts. I suggest that the authors review how to attach these figures, improving their resolution, for reading and evaluation. 4. Remove yellow highlights from some places in the article. 5. Review writing standards for the topic "reference"Author Response
Reviewer 1
The article “The challenges of partnering to promote health through sport” is very interesting and presents an important theme for the specific area. As a contribution to a better understanding of the work, some changes need to be made:
1. The authors separate “Introduction” and “Research context”, however, some points written in the “Research context” can be deleted, as they are not directly linked to the main topic of study, but to the historical context. In this way, I suggest that the authors summarize the information and remove the topic “Research context”, adding the short text to the “introduction”.
Thank you for this feedback. We have removed the ‘Research context’ section and revised the ‘Introduction’ to incorporate a shorter and more relevant version of the information we believe is important to enable the reader to understand the context within which the study was conducted.
2. Remove “Box 1”. I suggest that the information that is in “Box 1” be written in text format, in its proper place.
We have removed “Box 1” and incorporated the information from Box 1 into a short paragraph towards the end of the ‘Introduction’.
3. The resolution of figures 1 and 2 is very bad in the file, which prevents reading of some important parts. I suggest that the authors review how to attach these figures, improving their resolution, for reading and evaluation.
We have incorporated higher resolution versions of Figure 1 and Figure 2 into the revised manuscript.
4. Remove yellow highlights from some places in the article. 5. Review writing standards for the topic "reference"
We have removed all yellow highlights from the revised manuscript
Reviewer 2 Report
Dear Authors,
It is an interesting and well–written study. This manuscript adds to the literature by exploring the partnership–related challenges that RSAs are experiencing. This study clearly presents CM methodology, the conclusions and practical implications that are important globally. The text is prepared with attention to details and high awareness. The manuscript is easy to read, and the writing style is clear.
However, there is a recommendation for minor changes in this manuscript (outlined below):
Please include a new first paragraph to the discussion starting from the main idea of the study, then discuss the main findings and finally discuss what this study adds to the scientific knowledge globally. The first paragraph of the conclusions might be included as the first paragraph of the discussion.
Author Response
Reviewer 2
It is an interesting and well–written study. This manuscript adds to the literature by exploring the partnership–related challenges that RSAs are experiencing. This study clearly presents CM methodology, the conclusions and practical implications that are important globally. The text is prepared with attention to details and high awareness. The manuscript is easy to read, and the writing style is clear.
However, there is a recommendation for minor changes in this manuscript (outlined below):
5. Please include a new first paragraph to the discussion starting from the main idea of the study, then discuss the main findings and finally discuss what this study adds to the scientific knowledge globally. The first paragraph of the conclusions might be included as the first paragraph of the discussion.
Thank you for this suggestion. We have taken the first paragraph of the conclusion and incorporated it in to the first paragraph of the discussion along with an open sentence that re-states the aim of the study.